# Application of Bagging, Boosting and Stacking Ensemble and EasyEnsemble Methods for Landslide Susceptibility Mapping in the Three Gorges Reservoir Area of China

**DOI:** 10.3390/ijerph20064977

**Published:** 2023-03-11

**Authors:** Xueling Wu, Junyang Wang

**Affiliations:** School of Geophysics and Geomatics, China University of Geosciences, Wuhan 430074, China

**Keywords:** landslides, susceptibility, ensemble model, data balance, Three Gorges

## Abstract

Since the impoundment of the Three Gorges Reservoir area in 2003, the potential risks of geological disasters in the reservoir area have increased significantly, among which the hidden dangers of landslides are particularly prominent. To reduce casualties and damage, efficient and precise landslide susceptibility evaluation methods are important. Multiple ensemble models have been used to evaluate the susceptibility of the upper part of Badong County to landslides. In this study, EasyEnsemble technology was used to solve the imbalance between landslide and nonlandslide sample data. The extracted evaluation factors were input into three bagging, boosting, and stacking ensemble models for training, and landslide susceptibility mapping (LSM) was drawn. According to the importance analysis, the important factors affecting the occurrence of landslides are altitude, terrain surface texture (TST), distance to residences, distance to rivers and land use. The influences of different grid sizes on the susceptibility results were compared, and a larger grid was found to lead to the overfitting of the prediction results. Therefore, a 30 m grid was selected as the evaluation unit. The accuracy, area under the curve (AUC), recall rate, test set precision, and kappa coefficient of a multi-grained cascade forest (gcForest) model with the stacking method were 0.958, 0.991, 0.965, 0.946, and 0.91, respectively, which a significantly better than the values produced by the other models.

## 1. Introduction

As the most common geological disaster, landslides are harmful and destructive and can have a serious impact on human lives and the safety of public facilities [1]. A landslide is a disaster phenomenon in which a rock and soil mass on a slope, under the influence of natural conditions and human engineering activities, slides down the slope as a whole or scatters along the failure surface under the action of gravity. In such cases, if the slope mass is in an unstable state, the event may evolve into a landslide [2]. Landslide disasters have occurred frequently in the Three Gorges area. The Three Gorges Reservoir project has had large influences on the environment, geological disasters, society and the economy, and the region has received extensive research attention. More than 2500 slope failure sites are known in this area [3]; due to the construction of dams, the risk of landslides in the area has increased, and these landslides have huge potential risks. If an effective and accurate landslide susceptibility prediction system can be established, the extent of losses caused by landslide disasters will be minimized [4].

Landslide susceptibility evaluation is particularly important for the prediction and management of landslides. By analysing and quantifying the relationship between landslides and landslide-influencing factors, landslide-prone areas can be predicted to avoid deaths and economic losses caused by landslide disasters. In this paper, the landslide susceptibility in Badong County was evaluated using a data balance method and three ensemble model methods of bagging, boosting and stacking.

The occurrence of landslides is related to many environmental factors, and landslide susceptibility assessment explores the connections among them. Through investigations of landslide data from a project in 2020, a detailed landslide inventory map was established. Using correlation coefficient analysis, environmental factors were selected as independent variables. These environmental factors were extracted from digital elevation model (DEM) data, geological maps, Landsat-8 images, basic geographic databases and land cover data; the factors included profile, slope, aspect, altitude, slope length, slope height, slope pattern, plane curvature, middle slope location, terrain surface texture (TRI), terrain convergence index (TCI), terrain surface convexity (TSC), topographic position index (TPI), TST, valley depth, flow path length, catchment slope, distance to rivers, topographic wetness index (TWI), stream power index (SPI), land use, distance to roads, distance to residences, normalized difference vegetation index (NDVI), and structure data. Using the grid unit as the evaluation unit, the quantitative relationship between 25 landslide factors and landslide location was calculated by using representative models based on the three ensemble methods of bagging, boosting, and stacking: random forest (RF), extreme gradient boosting (XGBoost) and gcForest. Finally, the evaluation accuracy of landslide susceptibility was verified by comparing the AUC, test set precision, accuracy and recall rate for a known landslide.

In this paper, ArcGIS 10 software, SAGA-GIS software, PyCharm software, and the SPSS 20 statistical program were used for data processing, statistics, and mapping. The technical roadmap of this paper is shown in Figure 1.

## 2. Previous Work

Landslide susceptibility is evaluated by determining the combination of factors that have the greatest impact on the occurrence of landslides after detailed analysis of the landslide generation conditions; consequently, the possibility of landslides occurring in a given area can be estimated [5]. Economic development and the continuous expansion of the scope of human engineering activities have led to an increased impact of human beings on the environment; the number of landslide disasters has increased continually, and the resulting losses have become increasingly serious. Therefore, the use of efficient and reliable landslide hazard evaluation technology for landslide susceptibility evaluation is critical to quickly and accurately identify areas highly prone to landslide hazards and predicting the locations of new landslide hazards. This approach can provide efficient disaster forecasts and reduce the losses caused by landslide hazards. Auxiliary opinions can also be provided for the prevention of geological disasters. To study landslide susceptibility mapping, early researchers proposed various methods and techniques to improve the accuracy of landslide prediction.

Research on the susceptibility evaluation of landslide hazards began in the 1960s. Since the 1990s, mathematical statistics, probability theory, information theory, and fuzzy mathematics theory have been continually introduced into the field of geological disaster research. Traditional qualitative research has gradually moved towards quantitative research—that is, analyses based on data and information—to more objectively and scientifically reflect the true conditions of landslide geological disasters. At present, GIS-based methods for landslide geological hazard evaluation can be roughly divided into quantitative and qualitative evaluation approaches. With the continuous development of instruments and methods to obtain spatial data, the quality and quantity of spatial data have improved. Data-driven models, such as support vector machines (SVMs) [6], RF [7], artificial neural networks (ANNs) [8], and weight-of-evidence [9,10] models, have been used to produce regional LSM. In the data-driven model category, machine learning models provide a better prediction effect and higher accuracy than other approaches, such as expert-opinion-based methods and analytical methods [11]. SVM and ANN models are widely used in landslide mapping and generally yield good prediction results.

Although some machine learning methods perform well in terms of mathematics, explanations of the internal connections between landslide hazards and various factors remain unavailable. Before constructing an LSM, to analyse the effects of influencing factors on landslide occurrence, the mechanism of landslide occurrence must be fully understood, especially in areas threatened by different types of landslides [12]. Factor correlation analysis can eliminate the highly correlated factors that influence landslides, and importance analysis can be used to discern the effects of factors influencing landslides on landslide occurrence, thereby providing a powerful technical approach for selecting the important factors that influence landslides and performing landslide development trend analysis. However, a single learner is prone to underfitting or overfitting. To obtain a learner with high prediction accuracy and no overfitting, multiple individual learners can be combined through a certain combination strategy. This method of combining multiple individual learners is called ensemble learning. Some studies [13,14] have applied integrated models to landslide susceptibility modelling, but few researchers have compared and analysed the integration of three models for assessing landslide susceptibility. Some landslide susceptibility studies [15] used a variety of integrated models but did not consider the problem of landslide data imbalance or other study areas.

Landslide databases are usually unbalanced because landslide data are stored in raster format, and landslide grids are much less common than nonlandslide grids. The data imbalance problem affects the prediction performance of machine learning models by creating bias towards the dominant class. This problem can be overcome through data set balancing. At present, there are many studies on data balancing. The most common method is random sampling [7], which is used to make nonlandslide and landslide data sets roughly the same size. Agrawal et al. compared random oversampling and synthesized oversampling for landslide prediction [16]. As a new approach to correct imbalanced landslide datasets, a generative adversarial network (GAN) was tested at Chukha Dzongkhag, Bhutan. The results show that both the GAN and synthetic minority oversampling technique (SMOTE) data balancing approaches help improve the accuracy of machine learning models [17]. Compared with the above works, this study compares the prediction effects of integrated bagging, boosting and stacking models based on a landslide sensitivity assessment in Badong County, the Three Gorges region. Additionally, the EasyEnsemble method was used to deal with unbalanced sample data.

## 3. Study Area

The Three Gorges area was formed by the severe incision of lower Palaeozoic and Mesozoic massive limestone mountains (Jialinjiang Group, J1) along narrow fault zones in response to Quaternary uplift [18]. Steep slopes are widely developed on outcrops of erodible or “soft” materials, and landslides are common in these areas [19]. The Three Gorges region of the Yangtze River is in a mountainous gorge area where Sichuan and Hubei are connected. The area contains many mountains and steep slopes. In the event of heavy rain or earthquakes, disasters such as landslides, mudslides or rockslides easily occur. The study area is in Badong County (Figure 2). Located in the middle of Wu Gorge and Xiling Gorge in the Three Gorges region of the Yangtze River, Badong County is the area with the most complex geological conditions in the region. Folds and faults are widely distributed, and the geological structure is complex in this area. The whole Badong area has steep terrain, with a relative elevation up to 600 m.

## 4. Data Sources

Historical landslide catalogue data included information such as location, geological hazard body, area, and volume and were used to establish landslide distribution maps. Shuttle Radar Topography Mission (SRTM)1 DEM data with a spatial resolution of 30 m were used to extract topography and geomorphology information. Data acquired from the 1:250,000 national basic geographic database were used to determine the locations of residential areas, rivers, and roads. Bands 4 and 5 of the 2018 Landsat 8 image were used to obtain the NDVI. The 30-m global land cover data were the land use data. The NGAC-200,000 national geological map data provided information on the regional geological structure, strata, and lithology.

## 5. Primary Factors of Landslide Occurrence

In this paper, the main factors affecting the occurrence of landslides, such as topography, geomorphology, hydrological conditions, human engineering activities, surface cover, and basic geology, are discussed (Table 1). ArcGIS software and SAGA-GIS software were used to extract the following topographic factors from SRTM1 DEM data: profile, slope, aspect, altitude, slope length, slope height, slope pattern, plane curvature, middle slope location, TRI, TST, TPI, TSC, and TCI.

SAGA-GIS software was used to extract the valley depth, flow path length, catchment slope, distance to rivers, SPI, and TWI under hydrological conditions from SRTM1 DEM data. The distance to rivers, distance to residences, and distance to roads were obtained using 1:25 million national basic geographic databases to establish a buffer zone. The NDVI was obtained by calculations of the Landsat-8 image, the land use type was derived from the 30-metre global land cover data, and the geological structure was obtained from the geological map data. ArcGIS 10 software was used to extract the landslide impact factor layer and the landslide layer to the vector points and make them easy to analyse. The data set included 2,131,599 rows (number of grids) and 26 columns (25 factors and landslide data). Bivariate correlation analysis in SPSS software was used, and the Pearson correlation coefficient was used to calculate the correlation coefficient matrix of 25 landslide-influencing factors (Figure 3). Most of these 25 factors displayed low correlation coefficients, and the linear correlations between these factors were weak. Therefore, the 25 landslide impact factors were incorporated into the landslide susceptibility evaluation system to build the probability prediction model of landslide occurrence.

## 6. Method for Balancing Data Categories

The prediction of landslide disasters is a two-class problem in which the prediction results are only landslides or nonlandslides. An area should contain many more nonlandslide areas than landslide areas. Assuming that a landslide in the training data is grouped into class A and that a nonlandslide is grouped into class B, A:B = 1:99. In this case, if all samples in class A are classified as B, the error rate is only 1/100; however, if three samples in class B are classified as A, the error rate is 3/100. Achieving higher accuracy is the objective of most machine learning algorithms. The classification algorithms that strictly aim to maximize accuracy often ignore the correct classification of small samples, often leading to poor prediction results when processing unbalanced category samples [20].

In this case, the algorithm tended to predict all class A samples as class B samples. Landslide disasters are extremely harmful. If high-risk areas are classified as low-risk areas and a landslide occurs, it may cause many casualties and high economic losses. However, if low-risk areas are classified as high-risk areas, the loss is relatively small (generally, only an economic investment is made to prevent landslide disasters). Thus, the cost of misclassification of the two types of samples is different, and the spatial prediction of landslide disasters remains a cost-sensitive issue.

The problem of imbalance between the sample categories in landslide areas and nonlandslide areas can be solved at two levels: the algorithm level and the data level.

At the data level, the following three main data-level solutions are applicable: random sampling, SMOTE, and the EasyEnsemble method. For random sampling, to make the number of samples in the landslide and nonlandslide areas approximately the same, when selecting the training data set, the same amount of data was randomly sampled from landslide and nonlandslide areas. The important drawback of this scheme is that if the sample ratio was 1:10 and extraction without replacement was used, a maximum of 2 data points could be extracted, that is, a maximum of 2/11 data points could be included in the training set. This could lead to an insufficient training data set and make model training insufficient; consequently, the prediction accuracy could be low. In addition, if random sampling with replacement was used, the small-sample category would be repeatedly sampled many times, which may cause model overfitting, resulting in insufficient predictive ability.

The SMOTE algorithm can solve the overfitting problem in random sampling, and its core idea is to increase the quantity of data in low-sample categories to achieve data equalisation [21]. The new sample obtained by this method is related to the original sample and the neighbouring samples but retains important sample differences. This algorithm can improve the accuracy of landslide spatial prediction to a certain extent. However, this method is prone to overlap issues between new samples.

Another problem in random sampling is information loss. This problem can be solved using EasyEnsemble technology. EasyEnsemble technology is used to train a number of classifiers for ensemble learning by repeatedly combining positive samples with the same number of randomly sampled negative samples. This approach effectively solves the problem of unbalanced data types and reduces the loss of information due to undersampling. Therefore, EasyEnsemble technology was applied in this study to solve the problem of unbalanced sample types for landslide and nonlandslide samples. The technical process can be described as follows (Figure 4): (1) The entire training data set was divided into two categories, namely, majority and minority, which correspond to nonlandslide and landslide areas, respectively. (2) In each training, the nonlandslide area was randomly divided into n parts, and all samples in the landslide area were 1 part. (3) One piece was randomly selected from the nonlandslide sample to form a new training data subset, together with a landslide area. This subset was used to train the classifier, obtain the classification result and save it. (4) Steps (2) and (3) were repeated n times to obtain n classification results. (5) The average of the category scores of the n classification results was calculated to obtain the final classification result.

The solution at the algorithm level involves using the cost matrix to set the weights corresponding to different categories. The idea is that the cost of misclassification of different categories varies, and different categories are assigned different penalty coefficients in the algorithm. The purpose is to distinguish as few samples as possible.

## 7. Ensemble Model

Landslide susceptibility is evaluated by predicting the possibility of landslides in a certain area by selecting the most favourable combination of factors for landslide occurrence after analysing the landslide occurrence conditions. Many scholars have used landslide susceptibility evaluations to identify potential high-risk areas within a region and reduce the dangers of landslides, and they have obtained good results. A landslide susceptibility evaluation includes the division of evaluation units and the selection of evaluation factors. Choosing a suitable model is important for obtaining satisfactory prediction results for landslide susceptibility evaluation.

In 1962, the idea of ensemble learning began to appear. The first appearance of a cascading multiclassifier ensemble system was in a book by Sebestyen. Ensemble learning became broadly studied in the 1990s when Hansen et al. proposed a neural network ensemble model that used voting to integrate output results and obtain a better classifier than a single neural network. Bagging, boosting and stacking are three typical approaches in ensemble learning. By combining several machine learning algorithms into a meta-algorithm for prediction modelling, errors can be reduced, and satisfactory predictions can be obtained.

The bagging ensemble algorithm [22] is an ensemble learning algorithm in the field of machine learning that was originally proposed by Leo Breiman. The combination of the bagging integration algorithm and other algorithms can effectively enhance the prediction accuracy and stability of classification methods. The main content of the algorithm involves establishing a training set S of size N and evenly selecting n subsets S_i_ of size N from S with replacement (self-service sampling method) as a new training subset. By using these n training subsets, n training results can be obtained, and the analysis results are obtained through strategies such as averaging or voting (Figure 5). The main advantage is that formation learners that are not dependent on each other are generated in parallel. The bagging ensemble algorithm is suitable for prediction based on small-sample data sets and displays a good application effect in the field of machine learning.

The boosting algorithm [23] first uses the training set and initial weights to train weak learner 1. A weak learner refers to a learner with generalisation performance slightly better than that for a random guess [24]. Usually, different weights are assigned according to the classification accuracy, and the samples with low accuracy are given higher weights [25]. The samples with higher weights are considered by subsequent learners. Weak learner 2 is trained according to the training samples after adjusting the weights. The above steps are repeated t times to generate T base classifiers. The boosting framework algorithm assigns weights and fuses the N base classifiers to produce an improved classification result. After weighted fusion by the weak learners, the data are typically reweighted to strengthen the classification of previously misclassified data points. During the training of the boosting algorithm, the classifier is trained based on the samples with errors in the previous classification step, such that the algorithm can reduce the classification error rate of the model; as the training process progresses, the training set is increasingly correctly classified, and the variance of the model increases (Figure 6). However, the random sampling of features for training can reduce the correlation between submodels, thereby reducing the variance of the overall model [26].

In the stacking method [27], the primary learner is trained first, and then the prediction result of the primary learner is used as the new input to train the secondary learner. In the training phase, the secondary learner is generated by the primary learner. If the prediction results of the primary learner are directly used to generate the training set for the secondary learner, the risk of overfitting is high. Therefore, the initial training set is divided into k parts, and cross-validation is used to train each learner (Figure 7) [28].

Each of the three ensemble methods, bagging, boosting, and stacking, has multiple models. This article uses three representative models with the ensemble methods for landslide susceptibility prediction: an RF with bagging, the XGBoost model with boosting, and the gcForest model with stacking.

An RF is a classifier with multiple decision trees, each of which is a classifier [29]. After the decision trees are integrated, voting is used to determine the prediction result; that is, the prediction result is the category with the most votes. The random forest model is suitable for large-scale data prediction, especially for cases in which other models yield poor prediction results because of the high dimensionality of the sample. The accuracy of the RF model in most learning and prediction tasks can reach the same level as that of other models, and overfitting rarely occurs. The RF model has been widely used in competitions and practical applications. The model has two important parameters: the number of subtrees and the maximum number of features for a single decision tree.

The XGBoost algorithm is an improved method. The core idea of this algorithm is that multiple experts individually judge a complex task; then, the results are synthesized to reach a conclusion. The conclusion drawn after the synthesis is better than that provided by any one of the experts alone. The XGBoost algorithm is based on the regression tree model. The basic idea is to repeatedly extract certain variables to construct a regression tree model, obtain hundreds of regression tree models, and combine them linearly to obtain the final model.

The gcForest integration method is a new method based on decision tree forest aggregation. The gcForest integration method can make the data set of gcForest automatically learn its representation structure. The reason for this is that the method can automatically generate a decision tree forest with a higher-dimensional cascade structure. For example, when the decision tree input is a high-dimensional data set, the gcForest method can use a multigranular scanning method to increase the dimensional features, such that gcForest can effectively perform structural learning. In addition, in the gcForest method, the level of model complexity is automatically set, and the number of layers in the cascading forest is adaptively determined; consequently, the model can be trained with data sets of different sizes. In other words, gcForest automatically stops when the calculation result of the last cascade layer is lower than the expected value. Therefore, the gcForest method is suitable for both small-scale data and large-scale data training. In terms of the number of model parameters, the gcForest model has fewer parameters than ANN models, and it is also reliable for the parameter setting of the neural network.

## 8. Landslide Susceptibility Mapping

The grid unit was used as the evaluation unit in this study, and the multivalue extraction-to-point function in ArcGIS 10 software was used to extract 25 factors that influence landslides. The data set of the study area included 2,131,599 rows (number of grids) and 26 columns (25 factors and landslide data). With a 30 m grid, 269,421 pieces of data were labelled landslides, and 2,104,657 pieces of data were labelled nonlandslides; the ratio of landslide data to nonlandslide data was approximately 1:10. Therefore, we first randomly selected 25,000 landslide data and 205,000 nonlandslide data. Among the above data, we selected 5000 landslide data and 5000 nonlandslide data as test data and the remaining 20,000 landslide data and 200,000 nonlandslide data as training data. Extracting training data in this way can make the ratio of landslides to nonlandslides in the training data close to the actual ratio in the study area. Because the impact of grids of different sizes on landslide susceptibility needed to be compared, the data from 60 m and 90 m grids were processed similarly and organized into training and test sets. After an EasyEnsemble data balance was performed on the data set, the data were used to train the RF model with bagging, the XGBoost model with boosting, and the gcForest model with stacking. The training results were used to predict the probability of landslides for all samples from each model. The prediction results were added to the attribute table of the vector points in the study area, and then the vector point data were converted into raster data to create the maps of landslide-prone areas for the three models.

Feature importance measures the contribution of each input feature to the prediction results of a model and highlights the degree of correlation between a feature and the target. In this paper, the importance of 25 factors was calculated for the three tested models. The test results show that the altitude, TST, distance to residences, distance to rivers and land use are the main factors that affect landslide susceptibility (Figure 8).

The influence of altitude on the landslide distribution is mainly reflected in the local water collection platform caused by the topographic slope differences between different altitude ranges, the differences between the intensities of free surface and human engineering activities that are prone to landslides in different altitude ranges, and the characteristics of different vegetation types, coverages and atmospheric rainfall levels in different altitude ranges. Therefore, elevation is an important factor in landslide-prone environments. According to Table 2, the frequency ratio (FR) is greater than 1 in the altitude range from 49 m to 594 m. With increasing elevation, the frequency ratio decreases, which suggests that landslides are mainly distributed in low-elevation areas.

Terrain surface texture is one of the main parameters for representing the development characteristics of landforms. In places with complex terrain, such as ridges and valleys, the texture feature values are large, and in smooth and flat places, the texture values are small. According to Table 2, the frequency ratio of terrain surface texture values is greater than 1 in the range of 0.06 to 14.31, and the frequency ratio is largest in the range of 0.06 to 9.03, indicating that landslides in the study area are mostly distributed in areas with relatively smooth and flat terrain.

Human engineering activities generally involve resource exploitation and infrastructure construction processes based on certain engineering and technical measures, such as planning, design, construction, mining and operation. Human engineering activities can cause land erosion and change the original landform. Such activities can cause gradual and great harm. The areas where human engineering activities occur are often located near residential areas (examples include urban construction, irrigation activities, and traffic construction); thus, the distance from residential areas was used as an evaluation factor. According to Table 2, the corresponding FR is greater than 1 between 0 and 1040 m from a residential area, and the maximum ratio is within 614 m, indicating that the closer a residential area the activity is, the more likely it is that a landslide will occur.

The impact of rivers on landslide disasters in the study area mainly manifests in the lateral erosion and erosion-based cutting of bank slopes by river water. On the one hand, a river continuously creates higher and steeper bank slopes through erosion; on the other hand, it continuously washes the slope toe, causing the slope to always be in an unstable state. This process is important for the formation of new landslide masses and the revival of old landslide masses. Therefore, the distance from a river was selected as an evaluation factor to consider the impact of rivers on landslide disasters. Table 2 indicates that the FR is greater than 1 within the range of 451.15 m from a river, and the frequency ratio decreases with increasing distance from the river, indicating that landslides are more likely to occur in areas that are close to a river.

Land use refers to the long-term or periodic use, protection and transformation of land by using certain transformation means based on the natural attributes and characteristics of the land of interest. Five main types of land cover exist in the study area: cultivated land, forest, grassland, water bodies and artificial surfaces. According to Table 2, the regional FRs of artificial surfaces, cultivated land and water bodies are greater than 1 (especially the FR of artificial surfaces, which is the highest), and the FRs of grassland and forests are less than 1. Therefore, the landslides in the study area are more distributed in the areas where artificial surfaces, cultivated land and water bodies are located, and few landslides occur in forests and grasslands.

According to the landslide occurrence probability predicted by the model, a landslide susceptibility zoning map was created (Figure 9). The study area has five types of susceptibility levels: very low, low, medium, high, and very high. The RF model results are illustrated with the susceptibility map. Compared with other models, the RF model yields more extremely high landslide-prone areas and high landslide-prone areas. The gcForest model predicts the fewest extremely high and high landslide-prone areas. Most of the very low landslide-prone areas are in the southern and northern parts of the study area. The extremely high landslide-prone areas and high landslide-prone areas obtained with the three models are mainly located along the Yangtze River and in the middle and upper sections of the study area. The RF model identified many areas that were been labelled landslides in the past as high-susceptibility areas, such as the north bank in the western section of the Yangtze River in the study area. The XGBoost model generally predicted the locations where landslides occurred as high-susceptibility areas, and the gcForest model predicted a few areas to be more prone to landslides, while most landslides were located in highly prone areas.

The value obtained by dividing the landslide grid scale by the overall grid scale is the FR of the vulnerability grade. The higher the FR is, the greater the number of landslide grids per unit area at a given vulnerability level. Through the susceptibility results of the three models with different grid sizes and the statistical zoning table of landslides, the FR for each susceptibility level was calculated. The tables (see Table 3, Table 4 and Table 5 below) show that the higher the susceptibility level is, the higher the FR is, indicating that most landslides are in areas classified as highly and extremely highly prone to landslides, and the prediction results are reasonable. In the case in which the same model is used, with the gcForest model as an example, as the grid increases from 30 m to 90 m, the grid proportion for landslides in extremely highly prone areas changes little, from 97% to 88%; however, the grid proportion for highly prone areas increases by more than three times, from 3% to 10%. Therefore, the frequency ratio of highly prone areas decreases from 30.1380 to 8.9695, and the prediction effect of the model worsens. Therefore, a relatively small grid should be selected as the evaluation unit in studies of landslide susceptibility. When the grid size remains the same, the FR of the extremely highly prone areas identified with the RF model is the lowest, and that of the gcForest model is the highest, indicating that the gcForest model predicts fewer areas extremely highly prone to landslides but more areas that contain landslides; thus, its prediction effect is the best.

## 9. Validation of the Models

In an experiment comparing the influences of different grid sizes on the susceptibility results, the receiver operating characteristic (ROC) curves and AUC values of each model were obtained. The ROC curves and AUC values were calculated by using the probability obtained for the data predicted by the three models. The numbers of grids with different sizes varied. The number of grids with a 30 m grid size was 2,131,599, including 26,942 landslide grids. For a grid size of 60 m, the number of grids was 532,335, including 6715 landslide grids. At 90 m, the number of grids was 238,296, including 3009 landslide grids. Based on different grid sizes for the same model, the AUC value decreased with increasing grid size. The AUC value was largest for the 30-m grid and smallest for the 90 m grid. Based on different models with the same grid size, the AUC value of the gcForest model was highest, and that of the RF model was lowest, indicating that the prediction effect of the gcForest model was the best (Figure 10).

The effects of different grid cell sizes on the susceptibility results were assessed, and the larger the grid size was, the higher the accuracy of the training data and the lower the accuracy of the test data (Table 6). This result suggests that model overfitting occurs with increasing mesh size. As the grid becomes larger, the gap between the accuracy achieved for the training data and test data becomes larger, especially for the gcForest model. When the grid size was 90 m, the difference between the training data and test data accuracies of the gcForest model was as high as 15.2%. Therefore, in this paper, a 30 m grid was selected as the evaluation unit for landslide susceptibility modelling, such that high prediction accuracy could be obtained without overfitting.

The following Table 7 lists the prediction accuracies of the RF, XGBoost and gcForest models for samples in the study area. The AUC is an evaluation index used to measure the advantages and disadvantages of binary classification models. From the definition, the AUC can be obtained by summing the areas of each part under the ROC curve; its value represents the probability that a case is positively predicted. The recall rate indicates how many positive examples in the sample are predicted correctly. The accuracy is the number of samples correctly predicted for the positive class and accounts for the proportion of all positive samples predicted. The kappa coefficient can be used to test consistency and evaluate the accuracy of multiclass classification models. Whether the actual classification results of the model are consistent with the prediction results is the consistency of the classification problem. The kappa coefficient is obtained by calculating the confusion matrix, and it varies between −1 and 1 but is generally greater than 0. The accuracy, AUC value, recall rate, test set precision, and kappa coefficient of the gcForest model with stacking were 0.958, 0.991, 0.965, 0.946, and 0.91, respectively, which are significantly better than the values of the other two models.

## 10. Conclusions

This paper is a comparative study of multiple ensemble models of landslide susceptibility assessment in the upper half of Badong County in the Three Gorges area. Pham et al. noted that ensemble models provide excellent performance for future landslide prediction [30]. The landslide data were obtained from historical landslide records. In this landslide susceptibility analysis, 25 factors that influence landslides, including slope, aspect, plane curvature, profile curvature, elevation, and others, were used. According to the importance analysis, the important factors affecting the occurrence of landslides are the altitude, TST, distance to residences, distance to rivers and land use. The influences of different grid sizes on the susceptibility results were compared, and larger grids led to overfitting of the prediction results, as also reported in other studies [31]. Therefore, a 30 m grid was selected as the evaluation unit, and the study area contains 2,131,599 grid units. Due to the imbalance between the sample landslide data and the nonlandslide data, ensemble data balance processing was performed on the sample to construct the test data and the training data. The RF model with bagging, XGBoost model with boosting, and gcForest with stacking were used for training and prediction, and LSM were generated. According to LSM, the locations of the extremely high landslide-prone areas and high landslide-prone areas in the three models were generally consistent with the locations of historical landslides. The surrounding areas of the Yangtze River and its tributaries and the middle and upper areas of the study area are very prone to landslides.

The LSM was verified using the ROC results and known landslides. The quantitative results show that the order of the AUC values from small to large is RF model > XGBoost model > gcForest model. Additionally, the findings agree with those of Wei et al., who reported that the prediction accuracy of an XGBoost model was higher than that of an RF model [32]. The accuracy, AUC value, recall rate, test set precision, and kappa coefficient of the gcForest model with stacking were 0.958, 0.991, 0.965, 0.946, and 0.91, respectively, which are significantly better than the values of the other two models. In conclusion, the analysed results obtained from the study provide very useful information for engineers and planners involved in landslide hazard mitigation and infrastructure planning.

There were some deficiencies in this study. Affected by the basic data available, only point data for historical disasters were used, and the buffer zones established for analyses lacked precision; therefore, there will be errors in the process of vulnerability assessment. There was also an insufficient understanding of the vulnerability model, and there is still room for optimizing model parameters. In a follow-up study, more accurate landslide data should be obtained via high-resolution remote sensing and combined with landslide point data. It is necessary to deeply understand the model principle and parameters, obtain more optimized model parameters, and improve the accuracy of landslide susceptibility evaluation.

## Figures and Tables

**Figure 1 ijerph-20-04977-f001:**
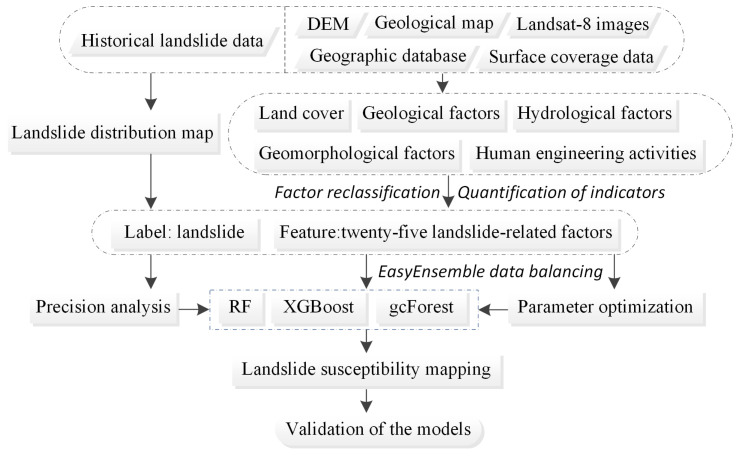
Flowchart of this study.

**Figure 2 ijerph-20-04977-f002:**
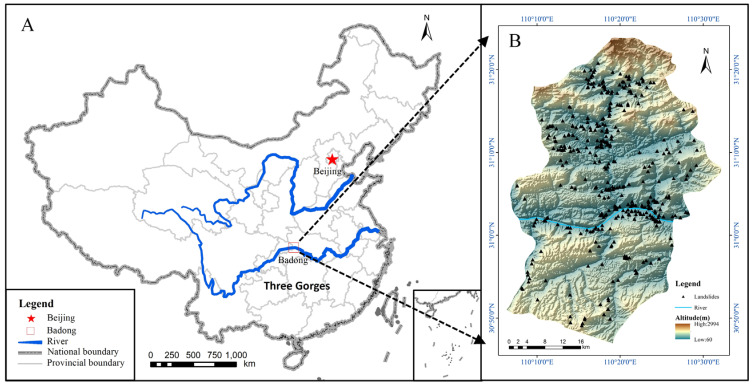
Location of the study area in China ((**A**) map of China, (**B**) map of Badong County).

**Figure 3 ijerph-20-04977-f003:**
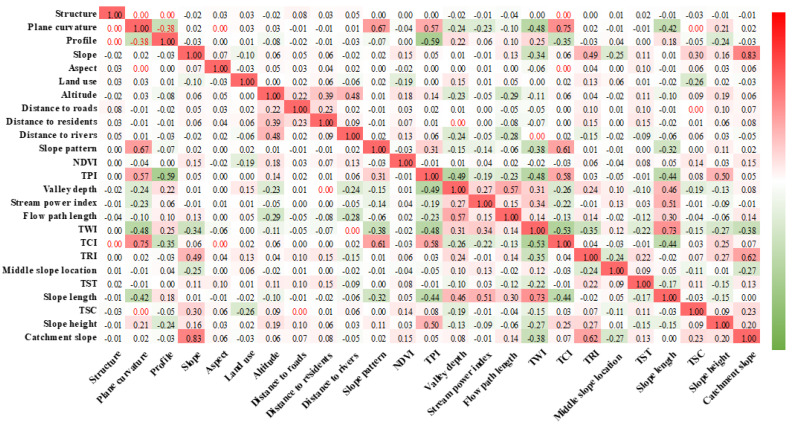
Correlation coefficient matrix of the causative factors of landslides (the black font indicates that the correlation coefficient *p* value is less than 0.05 and that the correlation is significant).

**Figure 4 ijerph-20-04977-f004:**
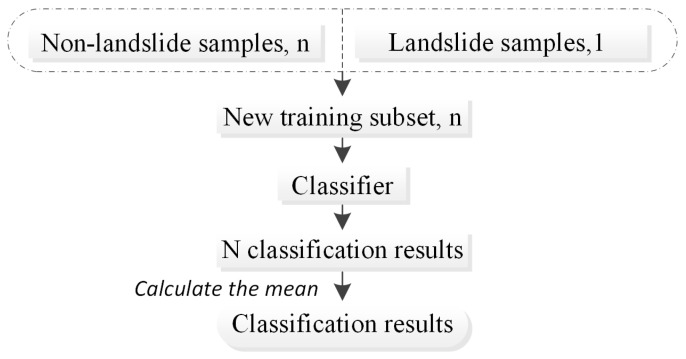
The EasyEnsemble technology approach can overcome the problem of unbalanced landslide samples.

**Figure 5 ijerph-20-04977-f005:**
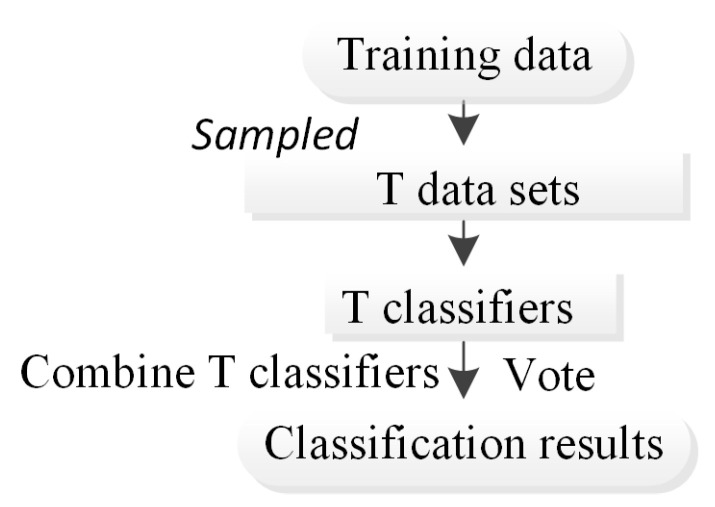
Flowchart of the bagging method.

**Figure 6 ijerph-20-04977-f006:**
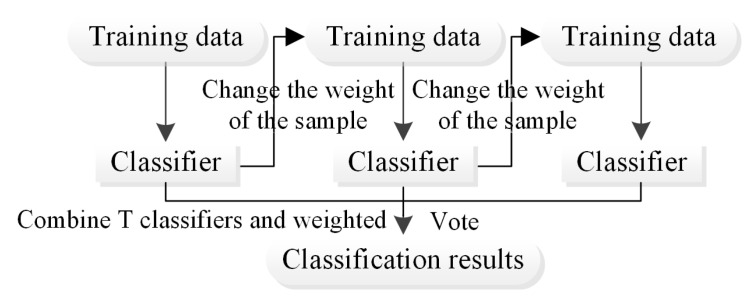
Flowchart of the boosting method.

**Figure 7 ijerph-20-04977-f007:**
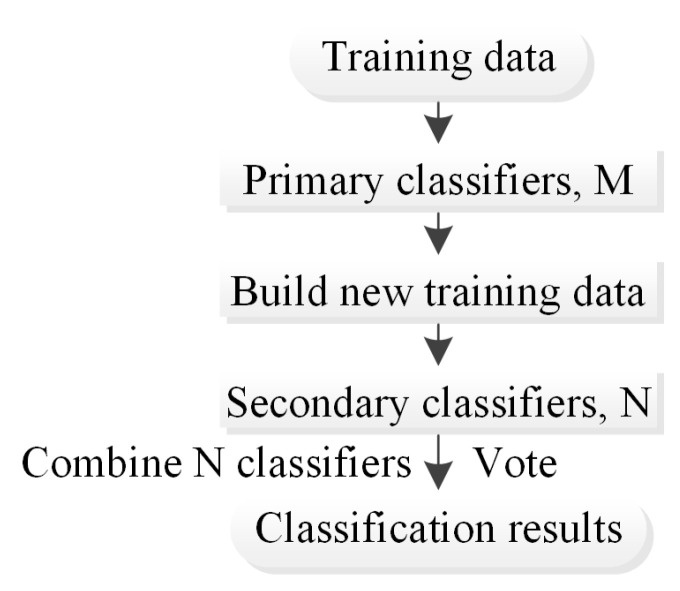
Flowchart of the stacking method.

**Figure 8 ijerph-20-04977-f008:**
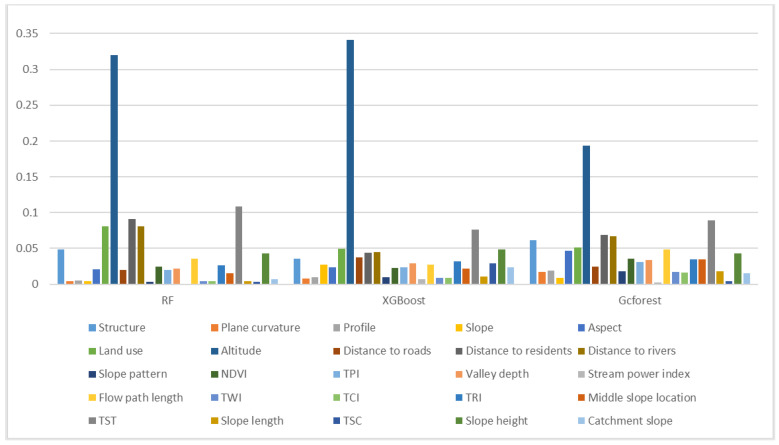
Feature importance measures (FIMs) for the factors that influence landslides in different models.

**Figure 9 ijerph-20-04977-f009:**
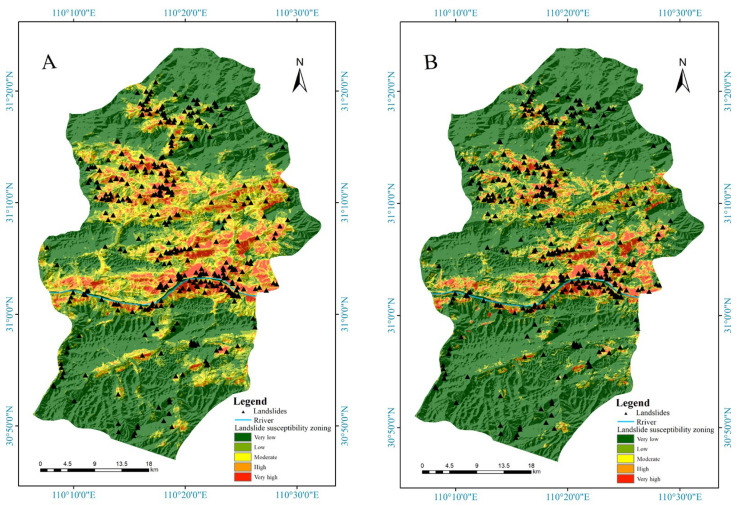
Landslide susceptibility zoning maps produced by three ensemble models: (**A**) RF model, (**B**) XGBoost model, and (**C**) gcForest model.

**Figure 10 ijerph-20-04977-f010:**
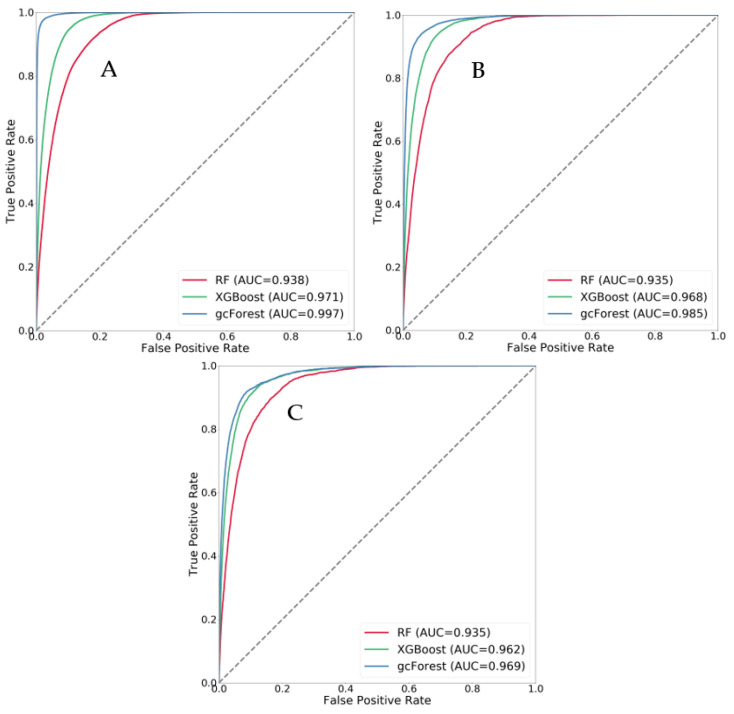
ROC curves of all data for different grid sizes ((**A**) 30 m, (**B**) 60 m, and (**C**) 90 m).

**Table 1 ijerph-20-04977-t001:** Descriptions of the causative factors of landslides.

Data Type	Factors	Source
Topographic features	Profile	DEM
	Slope	DEM
	Aspect	DEM
	Altitude	DEM
	Slope length	DEM
	Slope height	DEM
	Slope pattern	DEM
	Plane curvature	DEM
	Middle slope location	DEM
	TRI	DEM
	TST	DEM
	TPI	DEM
	TSC	DEM
	TCI	DEM
Hydrological conditions	Valley depth	DEM
	Flow path length	DEM
	Catchment slope	DEM
	Distance to rivers	GIS database
	SPI	DEM
	TWI	DEM
Human engineering activities	Land use	Surface coverage data
	Distance to roads	GIS database
	Distance to residences	GIS database
Surface cover	NDVI	Landsat-8 remote sensing images
Basic geology	Structure	Geological map

**Table 2 ijerph-20-04977-t002:** Statistical zoning table for the top five impact factors.

Evaluation Factors	Classification Level	Number of Pixels in Domain	Number of Landslides	Percentage of Domain	Percentage of Landslides	FR
Altitude	49–594	533,942	18,846	0.25	0.70	2.79
	594–937	579,737	6980	0.27	0.26	0.95
	937–1241	420,759	802	0.20	0.03	0.15
	1241–1561	349,147	333	0.16	0.01	0.08
	1561–1984	179,724	11	0.08	0.00	0.00
	1984–3096	67,119	0	0.03	0.00	0.00
Terrain surface texture level	0.06–9.03	287,378	9355	0.13	0.35	2.57
	9.03–14.31	456,151	8470	0.21	0.31	1.47
	14.31–18.88	500,238	5491	0.23	0.20	0.87
	18.88–23.28	433,011	2840	0.20	0.11	0.52
	23.28–28.21	313,383	659	0.15	0.02	0.17
	28.21–44.91	140,267	127	0.07	0.00	0.07
Distance to residences (m)	0–614.82	625,674	13,253	0.29	0.49	1.67
	614.82–1040.46	773,592	11,579	0.36	0.43	1.18
	1040.46–1489.75	473,118	1977	0.22	0.07	0.33
	1489.745–2104.56	174,421	133	0.08	0.00	0.06
	2104.56–3121.37	60,809	0	0.03	0.00	0.00
	3121.37–6029.93	22,547	0	0.01	0.00	0.00
Distance to rivers (m)	0–451.15	820,030	17,434	0.38	0.65	1.68
	451.15–1008.46	593,174	6963	0.28	0.26	0.93
	1008.46–1645.39	412,379	1909	0.19	0.07	0.37
	1645.39–2415.00	207,723	547	0.10	0.02	0.21
	2415.00–3529.62	75,110	89	0.04	0.00	0.09
	3529.62–6767.31	22,012	0	0.01	0.00	0.00
Land use	Cultivated land	580,187	16,364	0.27	0.61	2.23
	Forest	1,414,552	8388	0.66	0.31	0.47
	Grassland	93,621	813	0.04	0.03	0.69
	Water bodies	32,002	768	0.02	0.03	1.90
	Artificial surfaces	8823	605	0.00	0.02	5.42

**Table 3 ijerph-20-04977-t003:** RF zoning model of landslide susceptibility.

Grid Size	Landslide Susceptibility Level	Number of Pixels in Domain	Number of Landslides	Percentage of Domain	Percentage of Landslides	FR
30 m	Very low	1,153,876	38	0.54	0.00	0.0026
	Low	347,494	382	0.16	0.01	0.0869
	Moderate	260,594	2245	0.12	0.08	0.6812
	High	239,440	7737	0.11	0.29	2.5551
	Very high	128,884	16539	0.06	0.61	10.1469
60 m	Very low	219,759	5	0.41	0.00	0.0018
	Low	110,687	56	0.21	0.01	0.0400
	Moderate	77,065	342	0.15	0.05	0.3505
	High	70,236	1192	0.13	0.18	1.3403
	Very high	52,706	5122	0.10	0.76	7.6745
90 m	Very low	94,847	10	0.40	0.00	0.0083
	Low	48,629	44	0.20	0.01	0.0709
	Moderate	36,009	124	0.15	0.04	0.2700
	High	32,648	501	0.14	0.17	1.2031
	Very high	25,509	2352	0.11	0.78	7.2290

**Table 4 ijerph-20-04977-t004:** XGBoost zoning model of landslide susceptibility.

Grid Size	Landslide Susceptibility Level	Number of Pixels in Domain	Number of Landslides	Percentage of Domain	Percentage of Landslides	FR
30 m	Very low	1,544,816	80	0.73	0.00	0.0041
	Low	196,104	272	0.09	0.01	0.1097
	Moderate	145,685	1082	0.07	0.04	0.5872
	High	135,596	4738	0.06	0.18	2.7628
	Very high	108,089	20,770	0.05	0.77	15.1937
60 m	Very low	318,751	18	0.60	0.00	0.0045
	Low	71,661	52	0.14	0.01	0.0573
	Moderate	49,886	175	0.09	0.03	0.2770
	High	44,705	652	0.08	0.10	1.1516
	Very high	45,451	5821	0.09	0.87	10.1126
90 m	Very low	138,775	28	0.58	0.01	0.0158
	Low	32,145	61	0.14	0.02	0.1487
	Moderate	22,725	94	0.10	0.03	0.3242
	High	21,175	266	0.09	0.09	0.9846
	Very high	22,823	2583	0.10	0.85	8.8705

**Table 5 ijerph-20-04977-t005:** gcForest zoning model of landslide susceptibility.

Grid Size	Landslide Susceptibility Level	Number of Pixels in Domain	Number of Landslides	Percentage of Domain	Percentage of Landslides	FR
30 m	Very low	1,842,089	62	0.86	0.00	0.0027
	Low	105,676	104	0.05	0.00	0.0778
	Moderate	65,196	185	0.03	0.01	0.2244
	High	50,857	414	0.02	0.02	0.6437
	Very high	66,472	26,177	0.03	0.97	31.1380
60 m	Very low	328,806	24	0.62	0.00	0.0058
	Low	72,528	59	0.14	0.01	0.0642
	Moderate	49,390	132	0.09	0.02	0.2110
	High	40,401	385	0.08	0.06	0.7524
	Very high	39,329	6118	0.07	0.91	12.2830
90 m	Very low	118,010	16	0.50	0.01	0.0106
	Low	42,356	42	0.18	0.01	0.0777
	Moderate	28,854	92	0.12	0.03	0.2499
	High	25,179	222	0.11	0.07	0.6911
	Very high	23,244	2660	0.10	0.88	8.9695

**Table 6 ijerph-20-04977-t006:** Accuracies of the models with different grid sizes for training data and test data.

Model		30 m	60 m	90 m
RF	Train	0.873	0.890	0.912
	Test	0.862	0.851	0.838
XGBoost	Train	0.929	0.960	0.988
	Test	0.912	0.887	0.872
gcForest	Train	0.999	0.999	0.999
	Test	0.958	0.890	0.847

**Table 7 ijerph-20-04977-t007:** Statistical measures of different methods obtained for the training and test sets.

Data Set	Learning Method	Performance				
		Accuracy	AUC	Recall	Precision	Kappa
Training set	RF	0.873	0.943	0.933	0.808	0.749
	XGBoost	0.929	0.979	0.970	0.890	0.861
	gcForest	0.999	0.999	0.999	0.999	0.999
Test set	RF	0.862	0.932	0.914	0.805	0.725
	XGBoost	0.912	0.968	0.955	0.875	0.819
	gcForest	0.958	0.991	0.965	0.946	0.910

## Data Availability

The data that support the findings of this study are available from the corresponding author (X.W.) upon justifiable request.

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
