# Peer review of "Application of Bagging, Boosting and Stacking Ensemble and EasyEnsemble Methods for Landslide Susceptibility Mapping in the Three Gorges Reservoir Area of China"

_ijerph, 2023, doi:10.3390/ijerph20064977_

Round 1

Reviewer 1 Report

This manuscript compares the prediction effects of three classical ensemble models, namely, bagging, boosting and stacking, on the evaluation of landslide susceptibility in Badong County in the Three Gorges area. Besides, EasyEnsemble method is used to address unbalanced sample data. The research has a certain practical significance. However, paper writing and experimental discussion still need to be improved.

The reasons are as follows:

1.       In the introduction, please clarify the specific source of historical landslide data and the historical period of data collection (Line 48-50).   2.       In the previous work, only two studies in recent five years are cited and analyzed (Line 98 ‘Jayathissa et al. 2019’ and line 101 ‘Chowdhuri et al. 2021’), and the demonstration is insufficient.   3.       In the previous work, it is not mentioned whether other researchers use ensemble models to predict the landslide problem, and whether ensemble models in this manuscript are the first attempt to analyze landslide problem needs to be clarified.   4.       What analysis methods and functions are used to calculate correlation coefficients in SPSS 26 statistical software? (Line 165-166)   5.       In Figure 3, the correlation coefficient between catchment slope and slope is 0.83; The correlation coefficient between TCI and plane curvature is 0.75. Is the correlation between the above two groups of landslides influencing factors not strong?   6.       In Figure 3, please indicate the P value of the correlation coefficient and mark the correlation coefficient with significant difference.   7.       In line 333-337, the division of test data is not very clear. According to the author, 25,000 pieces of landslide data were randomly selected. 5,000 pieces of landslide data were removed, and the remaining 20,000 pieces of landslide data and were used as training data. Is there no landslide data in the test set? If so, the samples in the test set are all non-landslide data, and will face the problem of sample imbalance.   8.       Please briefly introduce the calculation formula of frequency ratio in this study and its significance.   9.       When discussing the analysis results involved in the Table, please quote the table directly rather than using long strings of words. (Line 361-365; line 369-372 and so on)   10.    What does FR stand for in Table 2-5? Please specify.   11.    If the content of these two columns (Percentage of domain (%) and Percentage of landslides (%)) is expressed in percentage, its value range should be 0-100. Therefore, 0.25(Percentage of domain when altitude is 49-594) should be represented by 25. So are the others.   12.    The Figure 10 is too vague in the manuscript. Please use clear picture.   13.    Please unify the decimal places of the values in Table 6 and Table 7.   14.    It is recommended to write the discussion and conclusion in more detail. According to the author's results, the prediction effect of the gcForest model in the stacking method is the best. Please analyze the possible reasons.  

Reviewer 2 Report

The paper "Application of bagging, boosting and stacking ensemble and EasyEnsemble methods to landslide susceptibility mapping in the Three Gorges Reservoir area of China" introduced methods for data imbalance in LSM. The introduction and literature are not good enough. You need to explain the problem of balancing data in the introduction, and in the literature, you need to mention the previous works and techniques which was used for data balancing. I suggest the author investigate more and enrich the introduction and the literature.  For example, GAN and SMOTE were used for such problems and you may refer to them. Reviewing the previous works and discussing the problems and the gap is necessary. Some recent works are mentioned in the following you may review. 

https://doi.org/10.3390/rs13194011

https://doi.org/10.3390/rs14133029

DOI: 10.1109/MLDS.2017.21

2- You need to explain the novelty and contribution clearly in the abstract and introduction. 

3-  You can move figure 1 to the methodology part  (workflow), and then describe the steps. 

4- Where are the maps of the influencing factors? I cannot see them. You should put them in your paper. 

5- The discussion part is poor and too limited; you need to extend it and discuss your significant findings and compare them with other similar works. Also, you need to mention bout the limitations of your work and mention your recommendations, as well as future work.

Round 2

Reviewer 2 Report

The paper has been improved. However, there are still some issues that need to be addressed. 

1- check the references and their location carefully.  Also, avoid merging too many references one after another. Try to mention what was the finding of each reference if you have it and then cite the work. 

For example: Agrawal K, Baweja Y, Dwivedi D, Saha R, Prasad P, Agrawal S, Kapoor S, Chaturvedi P, Mali N, Kala V.U (2017) A Comparison of Class Imbalance Techniques for Real-World Landslide Predictions. In Proceedings of the 2017 International Conference on Machine Learning and Data Science (MLDS). IEEE, Noida, India pp:1–8

 2-  The generation antagonism network (Al-Najjar HAH et al. 2021) has also been used to correct the class imbalance issue in landslide data sets.

  You need to mention briefly what was the main finding of the above-mentioned study. 

3- Regarding the previous comment in the first review, "Where are the maps of the influencing factors? I cannot see them. You should put them in your paper".

You may include the maps in the Appendix, or as extra materials. It is necessary for the reader to observe the factors and landslide locations for your mapping so one's can gain further ideas and insight.

4- Pham et al. noted that ensemble models provide excellent performance for future landslide prediction (Pham et al. 2020).

- Recheck the reference based on the standard of the journal. Where is the year from the above-mentioned reference?

5- English proofreading is suggested. 
